# Trapping Behaviour of *Duddingtonia flagrans* against Gastrointestinal Nematodes of Cattle under Year-Round Grazing Conditions

**DOI:** 10.3390/pathogens12030401

**Published:** 2023-03-01

**Authors:** Silvina Fernández, Sara Zegbi, Federica Sagües, Lucía Iglesias, Inés Guerrero, Carlos Saumell

**Affiliations:** 1Centro de Investigación Veterinaria de Tandil, UNCPBA-CICPBA CONICET, Tandil 7000, Buenos Aires, Argentina; 2Departamento de Sanidad Animal y Medicina Preventiva, Facultad de Ciencias Veterinarias, Universidad Nacional del Centro de la Provincia de Buenos Aires, Tandil 7000, Buenos Aires, Argentina; 3Centro de Investigaciones en Sanidad Animal, Pública y Ambiental, Facultad de Ciencias Veterinarias, Universidad Nacional del Centro de la Provincia de Buenos Aires, Tandil 7000, Buenos Aires, Argentina

**Keywords:** biological control, *Duddingtonia flagrans*, nematophagous fungi, gastrointestinal nematodes, infective larvae, cattle, livestock, integrated parasite control, parasite population

## Abstract

The purpose of using nematophagous fungi as biological control agents of gastrointestinal nematodes of livestock is to reduce the build-up of infective larvae on pasture and thus avoid clinical and subclinical disease. As the interaction of fungus-larval stages takes place in the environment, it is crucial to know how useful the fungal agents are throughout the seasons in areas where livestock graze all year-round. This study was designed to determine the predatory ability of the nematophagous fungus *Duddingtonia flagrans* against gastrointestinal nematodes of cattle during four experiments set up in different seasons. In each experiment, faeces containing eggs of gastrointestinal nematodes were mixed with 11,000 chlamydospores/g and deposited on pasture plots. A comparison between fungal-added faeces and control faeces without fungus were made with regard to pasture infectivity, larval presence in faecal pats, faecal cultures, faecal pat weight, and temperature inside the faecal mass. In three of the four experiments, *Duddingtonia flagrans* significantly reduced the population of infective larvae in cultures (68 to 97%), on herbage (80 to 100%), and inside the faecal pats (70 to 95%). The study demonstrated the possibility of counting on a biological control tool throughout most of the year in cattle regions with extensive grazing seasons.

## 1. Introduction

The search for and adoption of strategies for the control of gastrointestinal nematodes of cattle other than the use of anthelmintics has traditionally been justified by widespread anthelmintic resistance. Anthelmintic resistance has indisputably reached alarming levels in all major cattle production areas [1,2,3,4,5], making it unsustainable to rely solely on anthelmintics for the control of this disease. There are, however, other reasons to develop alternative control measures, which nevertheless are related to the massive use of chemical anthelmintics. Chief amongst these is the problem of anthelmintic residues and environmental contamination, which that are harmful in several ways (e.g., affecting non-target, beneficial organisms in soil, faeces, and water as well as pasture and vegetable plants) [6,7,8,9,10].

Alternative methods have therefore been designed and are the subject of numerous studies to date; these include genetic resistance, vaccines, nutraceutical compounds, and biological control by means of nematophagous fungi [11,12]. The purpose of using nematophagous fungi as biological control agents of gastrointestinal nematodes of livestock is to reduce the build-up of infective larvae on pasture and thus avoid clinical and subclinical disease [13]. This method is an appealingly eco-friendly approach given that it is innocuous to animals, residue-free, and has no negative environmental impact [14,15,16,17,18,19].

In most cases, the use of this tool has been confined to geographical areas with a reduced grazing season [20,21,22] or tropical/subtropical areas where seasonal changes throughout the year that can have a deep impact on fungal–nematode interactions are not well marked [23,24,25,26,27]. There have been very few reports where the main nematophagous fungus studied so far, *Duddingtonia flagrans*, has been tested in different seasons [28,29]. The challenge is to apply biological control in actual livestock production systems taking into consideration the various factors—environmental, parasitic, etc.—affecting the fungus–parasite interaction.

The present study aims to determine the ability of *D. flagrans* as a biological control agent of gastrointestinal nematodes of cattle during four seasons under year-round grazing conditions in the Humid Pampa rangelands of Argentina.

## 2. Materials and Methods

### 2.1. Experimental Site

The study took place on an experimental site of the Faculty of Veterinary Sciences in Tandil, Argentina (37°19′8.75″S, 59°04′39.8″W, Google Earth v.9.182.0.0). A newly mown, permanent, parasite-free pasture was used, which was composed of white clover (*Trifolium repens*), red clover (*T. pratense*), perennial ryegrass (*Lolium perenne*), and orchardgrass (*Dactylis glomerata*). The Köppen–Geiger climatic classification for this region is Cfa [30]. The climate is described as temperate, humid to sub-humid, with hot summers and frequent frosts in winter [31]. The historical average maximum and minimum temperatures for summer are 28 °C and 13 °C, respectively, while the same temperatures for winter are 13 °C and 1 °C. The historical average annual rain of 900 mm is distributed throughout the year, with the cold months receiving lower amounts of rain (less than 40%) [32].

### 2.2. Experimental Design

Four experiments were set up each during four distinct seasons (i.e., winter, spring, summer, and autumn). The general setup was the same for all experiments, with small differences due to seasonality (see below). Faeces were collected from calves belonging to an organically-run farm and harbouring natural infections with gastrointestinal nematodes. The faeces were thoroughly mixed before dividing them into two large batches: one batch was further mixed with a watery solution containing *D. flagrans* chlamydospores at a concentration of 11,000 chlamydospores per gram of faeces (cpg) (fungus group), and the second batch was mixed with the same volume of only water (control group). Eleven 500 g manually formed faecal pats from each batch were deposited on pasture. Each faecal pat was separated by a minimum of 1.5 m from the neighbouring ones; care was taken to avoid depositing the pats on any sloped area. Each pat was deposited onto a 15 × 15 cm nylon net of 10 mm mesh, which facilitated the collection of each e remaining pat at the end of each experiment while still allowing the normal faecal colonisation by micro- and macro-fauna. One faecal pat of each group was randomly selected as the “sentinel”; from these pats, a wedge sample of 40–60 g was taken weekly to detect the presence of infective larvae (L3) by baermmanisation, thus indicating the appropriate time to start the herbage sampling. Throughout each experiment, the temperature inside the faecal pats was recorded by electronic sensors (Datalogger Hygrochron iButton^®^, iButtonLink Technology, Whitewater, WI, USA) placed inside two randomly selected pats, one from each group at the time of deposition (day 0), except in the spring experiment, when only one sensor was used, placed inside one of the fungus-added faecal pats. Rainfall and air temperature data were obtained from the onsite weather station belonging to the Instituto de Hidrología de Llanuras “Dr. Eduardo J. Usunoff”, UNCPBA.

### 2.3. Faecal Cultures

Faeces from the same batch mixtures were used to set up ten faecal cultures each time faecal pats were deposited on pasture in order to corroborate the in vitro fungal predation as well as to determine the number and genera of the developed L3 after 14 days of incubation.

### 2.4. Herbage Sampling

A frame was used to mark off the sampling area around each faecal pat, which extended approximately 25–30 cm beyond the periphery of the pat [33]. The herbage within this area was collected twice during the spring and summer experiments by randomly selecting 2 × 1/4 of the sampling area, or once in its totality for the autumn and winter experiments (see below for differences due to seasonality). Herbage collection started 7–10 days after sufficient numbers of L3 were detected in the faecal pats.

### 2.5. Faecal Pats Recovering

At the time of the last herbage collection, the faeces remaining on the net were collected, weighed, and baermmanised to extract L3 for subsequent counting and identification.

### 2.6. Experimental Setup Differences Due to Seasonality

The differences in the length of each experiment relate to the eggs–L3 development time (i.e., the time lapsing between the deposition on the pasture of nematode eggs inside faecal pats and the first appearance of a minimum number of L3 in faeces) [34], which determined the timing of herbage sampling. The autumn and winter experiments had only one sampling time, while the spring and summer experiments had two samplings separated by 14 days. The setup of two samplings during the warmer months was to avoid the risk of eventual rains washing out L3 off the sampling area given that they appeared sooner on the grass than in the colder months. The winter experiment ran from 2 July to 25 August (54 days), the spring experiment ran from 1 to 20 November (28 days, with herbage samplings at day 15 and 28 post-deposition), the summer experiment ran from 24 February to 28 March (33 days, with samplings at day 20 and 33 post-deposition), and the autumn experiment ran from 24 April to 13 June (47 days).

### 2.7. Fungal Material

The local *D. flagrans* isolate 03/99 [35] was used. The fungus was grown on enriched Sabouraud agar cultures [36] at 27 °C for a minimum of 4 weeks before harvesting the chlamydospores by moistening the culture with distilled water and gently scraping off the agar surface using a stainless steel spatula. The fungal mass obtained was gently strained through a nylon mesh (100 µm mesh size) to break up the mycelia and release the individual chlamydospores. These were collected in glass beakers to which distilled water was added up to 40 mL. A 10 µL sample from this solution was further diluted in distilled water up to 1 mL, from which a second 10 µL aliquot was taken in order to count the chlamydospores under a light microscope. The chlamydospore suspensions were stored at 4 °C for up to 14 days before they were used. During the winter experiment, it was discovered that the chlamydospores used were part of a batch kept refrigerated for approximately one year, stored in tubes as a wet mass that had been subjected to temporary freezing. A quality control check of this fungal material was then performed while the experiment was running by incorporating chlamydospores and *Panagrellus* sp. to Petri dishes containing agar-agar that were incubated at 24 °C for 7 days; these cultures were inspected daily under a stereomicroscope for signals of fungal trapping behaviour.

### 2.8. Parasitological Procedures

Faecal egg counts (FEC) were determined using a modified McMaster method [37]. Faecal cultures were set up using a modification [37] of a standard technique [38] and incubated at 22–24 °C for 14 days. Recovery of L3 from the herbage samples was performed by herbage washing [39] and subsequent baermannisation. Washed herbage was placed in plastic net bags to dry off completely and then weighed. The extraction of L3 from faecal pats at the end of each experiment followed a baermannisation procedure designed to isolate L3 from the soil samples [40]. The L3 recovered from the faecal cultures, herbage, and faecal pats were stored at 5 °C until counted and identified under a light microscope using existing keys [41,42].

### 2.9. Statistical Analysis

One-way ANOVA followed by Tukey’s multiple comparisons test was performed to analyse the normally distributed data of the temperature inside the pats, while the non-normally distributed data were analysed using the Friedman test followed by Dunn’s multiple comparison test. Data of the faecal pat weights and numbers of L3 recovered from the faecal cultures, herbage, and faecal pats were analysed using the Mann–Whitney U-test for non-normal distributions or the unpaired *t* test with Welch’s correction for normal distributions. When data transformation was needed, the function Y = sqrt(Y) was used. Data analysis was performed using the software GraphPad Prism 9.4.1 for Windows, GraphPad Software, San Diego, CA, USA, www.graphpad.com.

## 3. Results

The FEC in the freshly deposited faecal pats was 405 eggs per gram (epg) in winter, 307 epg in spring, 160 epg in summer, and 590 epg in autumn. These resulted in an egg:chlamydospore (E:C) ratio of 1:27, 1:36, 1:69, and 1:19 for the faeces used in those seasons, respectively. The faecal cultures (Figure 1) revealed that the fungus was responsible for larval reductions of 11.8% (*p* = 0.4746) in winter, 67.7% (*p* = 0.0150) in spring, 86.2% (*p* < 0.0001) in summer, and 96.5% (*p* = 0.0002) in autumn. Table 1 shows the relative composition of the parasite population in faeces; *Ostertagia*, *Haemonchus*, *Trichostrongylus*, *Cooperia*, and *Oesophagostomum* were the genera present at the time of deposition on pasture, but the last genus was not present in the winter cultures. There were no differences regarding the relative larval composition of each genus between the control and fungus-added cultures.

The pasture infectivity around the faecal pats containing *D. flagrans* (Figure 2) was reduced by 23.9% (*p* = 0.2799) in winter, 80% (*p* = 0.0412) and 89.5% (*p* = 0.0007) (first and second sampling, respectively) in spring, 100% (*p* = 0.0325 and *p* = 0.0294) for both samplings in summer, and 91% (*p* < 0.0001) in autumn. The cumulative larval numbers (i.e., the sum of L3 from the two samplings in spring and summer) also showed a significant reduction in the fungus-added group of 83.9% (*p* = 0.0003) and 100% (*p* = 0.0007), respectively, compared to the control group. The parasite genera identified from the pasture samples in every season were *Ostertagia*, *Haemonchus*, *Trichostrongylus*, and *Cooperia*.

Varying degrees of the predatory effect of *D. flagrans* was observed when the faecal pats were recovered at the end of each experiment and the remaining L3 inside were extracted (Figure 3). A lower effect of the fungus was recorded in winter (20.9%, *p* = 0.7394) and summer (25.7%, *p* = 0.3180), while the fungal efficacy was much higher in spring (69.6%, *p* = 0.2915) and autumn (94.9%, *p* = 0.0004).

Figure 4 shows the weight of the recovered faecal pats in the four experiments. There were no differences in the final weight between the control and fungus-added faecal pats. After 54, 28, 33, and 47 days on pasture, the faecal pats lost 55%, 75%, 79%, and 48% of their fresh weight during the winter, spring, summer and autumn experiments, respectively.

The relative composition (Table 2) of the parasitic larval population remaining in the faecal pats after four to eight weeks (depending on the season) in the environment comprised the genera *Ostertagia*, *Haemonchus*, *Trichostrongylus*, *Cooperia*, and *Oesophagostomum* (this last one only being present in spring and summer). No differences were detected between control and fungus-added faecal pats regarding the relative composition of each genus. The numbers of L3 recovered from the faecal pats in each season as a proportion of the estimated number of eggs deposited on pasture for the control and fungus-added groups, respectively, were as follows: winter, 2.05% ± 1.45% and 1.62% ± 1.41%; spring, 17.16% ± 33.57% and 4.69% ± 4.89%; summer, 0.13% ± 0.09% and 0.12% ± 0.07%; autumn, 1.71% ± 0.93% and 0.09% ± 0.06%.

Rainfall and the maximum and minimum air temperature as well as the maximum and minimum temperatures inside the faecal pats are shown in Figure 5. No significant differences (*p* = 0.622 to *p* > 0.9999) were detected for the maximum or minimum temperatures inside the faecal pats between the control and fungus-added groups in the winter, summer, and autumn. The same comparative analysis could not be performed for the spring experiment, as the temperature was only recorded inside a fungus-added pat. The minimum air temperature during the four experiments showed a very similar pattern and corresponded closely to those recorded inside the faeces, although some differences were detected for the control group in winter (*p* = 0.0416), and for the fungus-added group in summer (*p* = 0.0236) and autumn (*p* = 0.0053). On the other hand, the maximum air temperature resembled the maximum temperature inside the faecal pats only in winter and the last days of autumn, while during spring, summer, the first half of autumn, and the last days of winter, the maximum air temperature was consistently and significantly lower than the temperature inside the faecal pats. These differences were detected for the control group during the last week of winter (*p* = 0.0086); the fungus-added group in spring (*p* < 0.0001); the control (*p* = 0.0030) and fungus-added (*p* < 0.0001) groups during summer and the control (*p* = 0.0323) and the fungus-added (*p* = 0.0397) groups during the first half of autumn. Malfunctioning of the air temperature recording equipment during the autumn experiment from 24 May to 8 June led to missing data for that parameter, therefore, the data analysis for this season did not include those sixteen days.

## 4. Discussion

This study confirmed the adaptable predatory capacity of *D. flagrans* on the free-living larvae of gastrointestinal nematodes of cattle under diverse environmental conditions such as the prevalent ones during year-round cattle grazing in the Humid Pampa’s region of Argentina. Previous reports of this type of studiy have been confined to two seasons (either winter and summer [28] or autumn and spring [29]). In the present study, the fungal predatory activity was measured during each of the four distinct seasons. The data analysis showed that from spring to autumn, the fungus could significantly reduce the number of L3 on pasture.

The quality control check performed while the winter experiment was running led to the conclusion that the fungus had not retained its high predatory capacity, probably due to the inactivation of chlamydospores kept as a wet mass under temporary freezing storage conditions. This is in agreement with a previous report describing a drastic fall in the germination rates of *D. flagrans* after liquid incubation for up to 6 months [43]. Therefore, the mix of fungal material and faeces deposited on pasture would have contained much fewer chlamydospores able to germinate than those stipulated by the experimental protocol. The poor performance of the fungus observed in the in vitro results from the faecal cultures certainly supports this conclusion. For this reason, comparisons of the winter results with a previous study [28] could not be made. Thus, the question as to whether *D. flagrans* can be used as a biological control agent during the winter season under local grazing conditions remains unanswered and deserves further exploration, especially considering the importance of winter gastrointestinal nematodosis in grazing cattle [44].

The inclusion of “sentinel” faecal pats to detect the appropriate time for herbage sampling in each experiment proved to be useful as the development of L3 in cattle faeces varies with the season. Under local conditions, the egg to L3 development takes 4–6 weeks in winter, 1–4 weeks in spring, 1–2 weeks in summer, and 3–5 weeks in autumn [34].

The use of temperature sensors inside the faecal pats was also extremely useful as it revealed just how much the air temperature (normally recorded at 2 m above soil surface) differed from the real temperature to which the micro- and macro-fauna are exposed inside the faecal mass, especially in the warmer months. This makes for a better understanding of the behavioural responses such as development, predation, survival, etc. from both the fungi and nematode larvae to fluctuating and very high temperatures, which has been noted in a previous summer trial [45].

Taking all of the studied ‘outdoors’ parameters as a whole, the results from the summer experiment might be considered as somewhat puzzling. On one hand, the high predatory effect of *D. flagrans* on gastrointestinal L3 was clearly evident on the pasture infectivity; on the other hand, the same degree of larval reductions was not observed in the faecal pats at the end of the experiment. Compared to the other seasons, the numbers of L3 in the control group transmitted to the surrounding herbage in summer was much lower, which could be attributed to two factors; lower numbers of eggs deposited on pasture in summer than in the other seasons, and the scarce rainfall of 44 mm during that experiment, much lower than the historical 106 mm for that time of the year [32]. The initial rainfall of 24 mm on day 1 of the experiment cannot be considered as an inducive factor for larval transmission because there was not enough time for the eggs to develop to L3, which takes 1–2 weeks under local conditions [34]. Perhaps the reason for low fungal effect related to L3 remaining in faecal pats lies in the combination of the environmental conditions (i.e., dryness (as explained above) and high temperatures) at which the faeces were subjected at that time. Although it has been shown that temperatures of ≥35 °C negatively affect both the growth and nematophagous capacity of *D. flagrans* in the Northern Hemisphere [46,47,48], records from the Southern Hemisphere have shown that the fungus is able to reduce larval populations at temperatures temporary reaching 39 °C and higher [49,50]. As the records of the summer experiment revealed that the temperature inside the fungus-added faecal pats surpassed 35 °C for as much as 3.5 to 5.5 h/day in 18 of the 33 days that the experiment lasted, it is possible to concur with previous authors [48,49] that the development and trapping efficacy of *D. flagrans* depend not so much on the high temperature limits, but rather on the duration of exposure to those limits.

Overall, and barring the winter study, which requires further attention, spring and autumn were the seasons when the reducing effect of *D. flagrans* on the gastrointestinal nematodes could be seen in full. All parameters studied (pasture larval infectivity, L3 remaining in faecal pats and faecal cultures as in vitro corroboration) showed very high fungal trapping activity in these seasons, despite the fact that the E:C ratios used in each season were different. The high larval reductions obtained in spring, summer, and autumn in both the faecal cultures and pasture infectivity are in agreement with previous reports demonstrating that large variations in FEC in cattle—and, consequently, variations in the E:C ratio at the moment of administering fungal chlamydospores—do not present an obstacle for the efficacy of *D. flagrans* on cattle nematodes [51].

The >90% reduction in the larval population by *D. flagrans* in autumn has important implications for the practical adoption of this biological control tool against gastrointestinal nematodes in cattle. Under local grazing conditions, weaning takes place in late-summer/early-autumn. It is at that time that the newly-weaned calves, grazing on pastures infected with over-summering L3, start recycling the disease, which then produces three to four generations of parasites by the time the herbage availability is at its natural lowest point in winter [44]. Therefore, using nematophagous fungi in autumn, either as a stand-alone strategy or combined with other parasite control strategies, would prevent the build-up of L3 on pasture and avoid the pernicious consequences of gastrointestinal nematodosis.

This study highlights the importance of recognising that successful implementation of a biological control using nematophagous fungi involves the understanding of biotic and abiotic factors influencing not only the survival and transmission of free-living stages in the environment [34,52,53], but also the fungal development and predatory ability [54,55]. The results obtained clearly emphasise the fact that *D. flagrans* not only tolerates an ample range of fluctuating temperatures in the environment, but can also successfully prey on the free-living larvae of gastrointestinal nematodes.

## 5. Conclusions

The results from the present study confirmed that *D. flagrans* is able to act as a biological control agent against gastrointestinal nematodes of cattle under most of the prevailing year-round grazing conditions in the Humid Pampa’s region of Argentina. The optimal results obtained in autumn shed light on the usefulness of incorporating biological control into the production cycle of grazing cattle.

## Figures and Tables

**Figure 1 pathogens-12-00401-f001:**
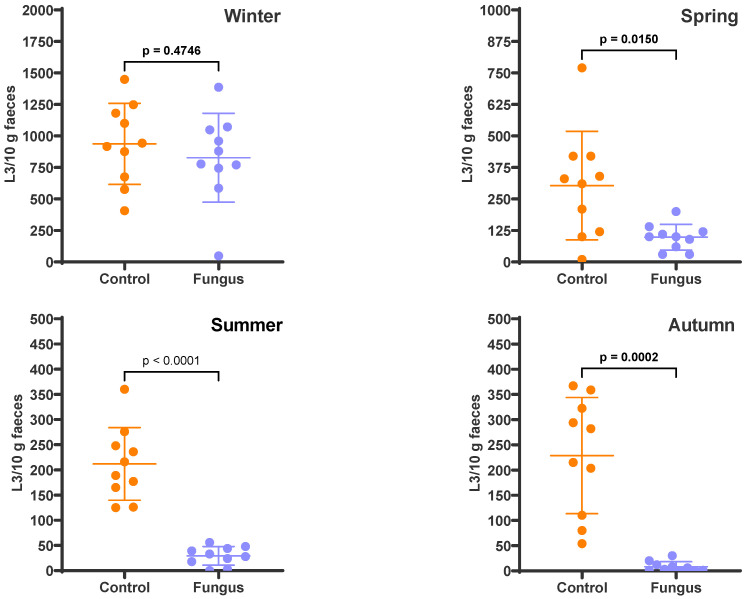
Numbers of infective larvae recovered from the faecal cultures, expressed as L3/10 g faeces. The scatter plot shows the mean (*n* = 10) with standard deviation, with each dot representing an individual value. Faecal cultures were set up in each season from cattle faeces containing eggs of gastrointestinal nematodes, with (Fungus) and without (Control) the addition of *D. flagrans* chlamydospores.

**Figure 2 pathogens-12-00401-f002:**
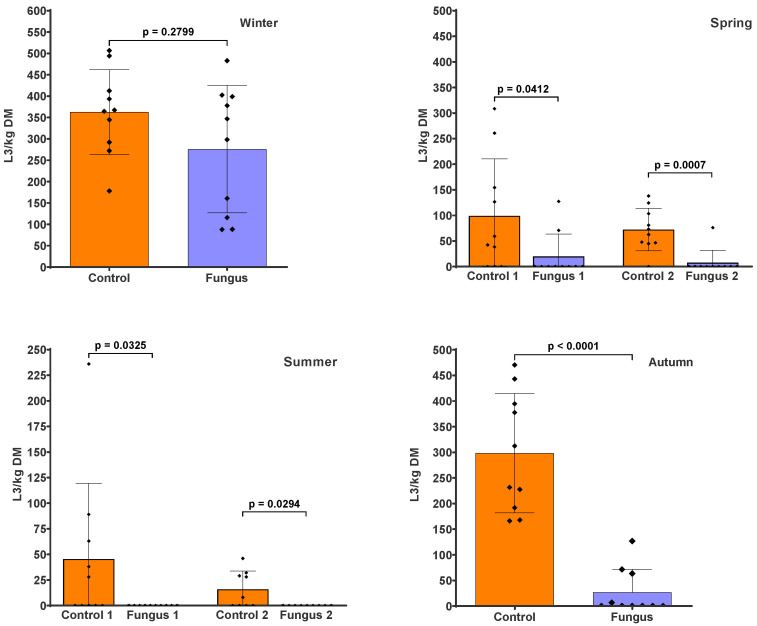
Numbers of infective larvae recovered from herbage around faecal pats with (Fungus) and without (Control) the addition of *D. flagrans* chlamydospores, expressed as L3/kg dry matter (DM). The scatter plot shows the mean (*n* = 10) with the standard deviation, with each dot representing an individual value. Herbage was collected at the end of the winter and autumn experiments, and twice during the spring and summer experiments.

**Figure 3 pathogens-12-00401-f003:**
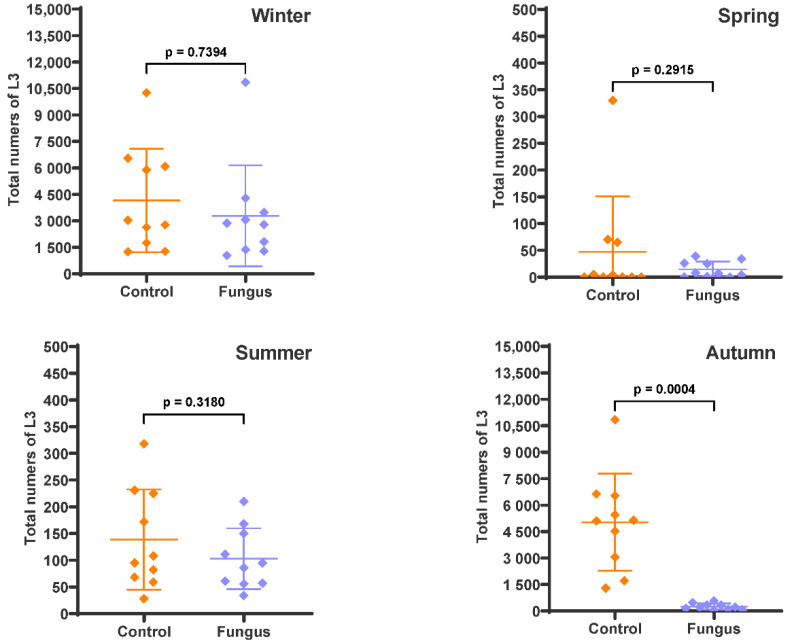
Numbers of infective larvae in the faecal pats expressed as the total numbers of L3. The scatter plot shows the mean (*n* = 10) with standard deviation, with each dot representing an individual value. Faecal pats containing eggs of gastrointestinal nematodes with (Fungus) and without (Control) chlamydospores of *D. flagrans* were deposited on pasture on four occasions and remaining L3 were extracted after 54 days (winter), 28 days (spring), 33 days (summer), and 47 days (autumn).

**Figure 4 pathogens-12-00401-f004:**
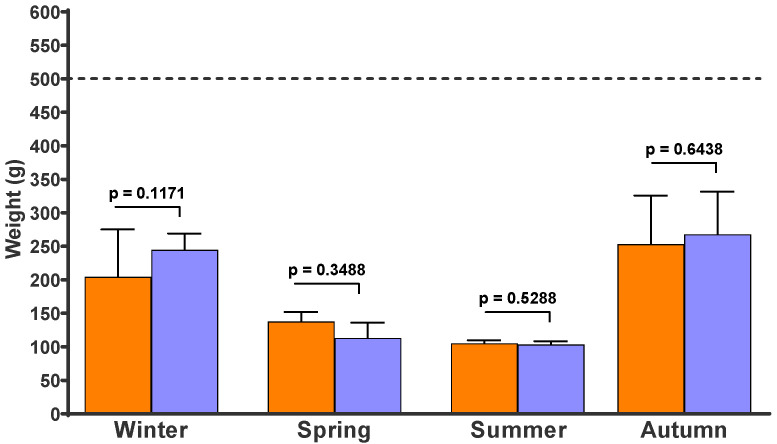
Weight expressed in grams (g) of faecal pats that had been on pasture for 54 days (winter), 28 days (spring), 33 days (summer), and 47 days (autumn). Each column represents the mean of 10 pats and its standard deviation. Red columns: Control faecal pats without chlamydospores of *D. flagrans*. Green columns: Fungus-added faecal pats. Dotted line: initial weight of freshly deposited faecal pats.

**Figure 5 pathogens-12-00401-f005:**
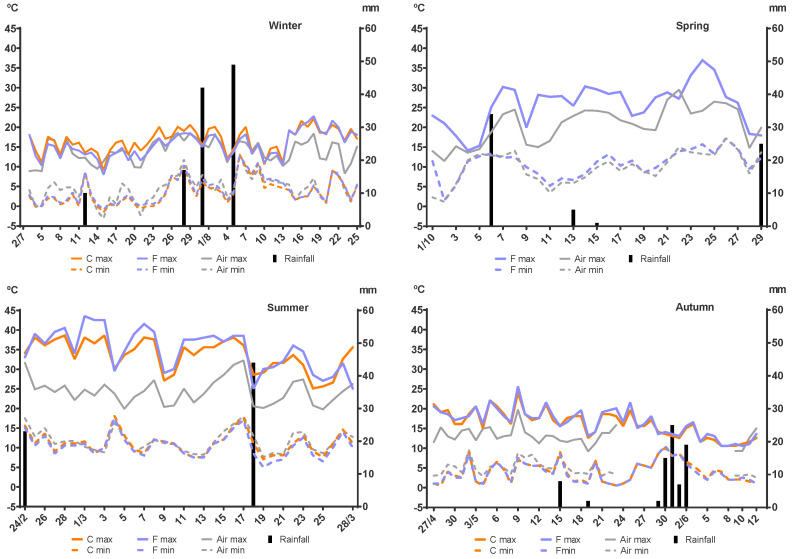
Daily maximum and minimum temperatures (°C) and rainfall (mm) recorded during each experiment. Red lines: temperature recorded inside a control faecal pat (not in spring). Green lines: temperature recorded inside a fungus-added faecal pat. Grey lines: air temperature recorded at 2 m above soil surface. Black columns: rainfall.

**Table 1 pathogens-12-00401-t001:** Genera of the gastrointestinal nematodes in the faecal cultures set up for the experiments carried out in different seasons. Each number represents the average percentage of ten cultures and the standard deviations are shown in brackets. Fungus = cultures set up with the addition of *D. flagrans* chlamydospores; Control = cultures without fungal chlamydospores.

	Winter		Spring		Summer		Autumn	
	Control	Fungus	Control	Fungus	Control	Fungus	Control	Fungus
*Ostertagia*	33.1 (8.452)	33.6 (4.695)	63.1 (16.84)	68.44 (15.07)	48.8 (9.762)	49.56 (21.93)	31.92 (8.835)	54.29 (34.83)
*Haemonchus*	13.7 (3.234)	11.4 (4.624)	16.64 (11.54)	4.74 (6.888)	15.2 (10.46)	22.56 (16.91)	18.56 (5.688)	16.55 (21.33)
*Trichostrongylus*	22.5 (7.487)	22.2 (15.01)	16.57 (10.04)	5.37 (5.852)	21.2 (6.812)	11.78 (16.0)	18.75 (10.65)	13.44 (12.30)
*Cooperia*	30.7 (3.622)	32.8 (13.34)	2.55 (3.951)	19.9 (13.21)	10.8 (4.237)	4.33 (5.431)	22.95 (9.555)	13.13 (16.77)
*Oesophagostomum*	0	0	1.15 (1.498)	1.55 (3.259)	4.0 (3.266)	11.78 (13.26)	7.82 (3.297)	2.60 (4.943)

**Table 2 pathogens-12-00401-t002:** Genera of the gastrointestinal nematode L3 recovered from faecal pats after 54, 28, 33, and 47 days post-deposition on pasture in winter, spring, summer, and autumn, respectively. Each number represents the average percentage of ten faecal pats, standard deviations are shown in brackets. Fungus = faecal pats with added *D. flagrans*; Control = faecal pats without *D. flagrans*.

	Winter		Spring		Summer		Autumn	
	Control	Fungus	Control	Fungus	Control	Fungus	Control	Fungus
*Ostertagia*	47.5 (5.206)	48.63 (9.086)	37.02 (23.33)	52.19 (27.00)	41.5 (12.01)	53.16 (13.23)	29.72 (5.414)	35.65 (15.03)
*Haemonchus*	3.67 (3.445)	3.5 (2.976)	17.96 (11.76)	10.29 (11.63)	18.64 (12.65)	10.85 (9.386)	16.26 (3.094)	11.39 (11.20)
*Trichostrongylus*	4.0 (4.899)	9.13 (4.704)	4.98 (6.943)	6.59 (6.539)	18.12 (12.67)	19.75 (14.97)	23.51 (8.352)	15.63 (10.87)
*Cooperia*	28.5 (4.461)	36.75 (5.339)	31.24 (17.71)	29.96 (14.71)	12.42 (11.53)	7.92 (7.883)	30.51 (5.616)	37.33 (16.76)
*Oesophagostomum*	0	0	5.72 (8.501)	0.98 (1.806)	9.3 (8.830)	8.32 (7.108)	0	0

## Data Availability

Not applicable.

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
