# Peer review of "Trapping Behaviour of Duddingtonia flagrans against Gastrointestinal Nematodes of Cattle under Year-Round Grazing Conditions"

_pathogens, 2023, doi:10.3390/pathogens12030401_

Round 1

Reviewer 1 Report

This document addresses a very interesting and practical topic, the usefulness of a biological control agent to reduce the risk of infection by gastrointestinal nematodes among grazing cattle. The investigation has been well designed and performed, congratulations to the authors. There are several points needing more attention:

1) The area where the experiments were performed is not described conveniently. More information regarding the geograpical localization is needed. There is a lack of data on the climatic conditions during each season. It is suggested that authors provide a climatic classification, i.e., Köppen classification, to allow that data collected can be extrapolated or compared with those from other areas in different countries.

2) In general, data should be provided as an average and a dispersion measure (SD, range...).

3) Is there any explanation for the highest numbers of L3s developed in winter? The differences in the counts of eggs in the fecal pats appears not clear enough. Please, explain.

4) The explanation about the chlamydospores used in winter should be provided in the Materials section, and not in the Discussion. Additionally, more detailed information is essential regarding the culturing and collection of chlamydospores, time elapsed...

5) A brief paragraph is needed in the Discussion regarding the quantity of chlamydospores added to the fecal pats. Besides this, the possibilities of such number of chlamydospores attain the fecal pat should be discussed. Would be the authors able to indicate an approximate amount of chlamydospores that cattle should ingest to ensure that al least a number of 11000 chlamydospores did attain the feces?

Reviewer 2 Report

The authors determined the predatory ability of the nematophagous fungus Duddingtonia flagrans against gastrointestinal nematodes of cattle during four experiments set up in different seasons. On the whole, I think the paper could merit publication in Pathogens, but after major revision.

1.     How the parasite population in faeces: Ostertagia, Haemonchus, Trichostrongylus, Cooperia and Oesophagostomum were identified? And is there any differenes between trapping behaviour of D. flagrans on the different nematodes?

2.     The daily maximum and minimum temperatures and rainfall varies in different seasons, is there any relationships between the indexes and predatory effects of the fungus?

3.     The figures use green and red as contrast colours, others are suggested to make red-green colour-blindness easy to read.

Round 2

Reviewer 2 Report

The revised version has improved a lot in both the language and presentation of results. And questions from the reviewer have been properly addressed.